# Dynamic Negative Guidance of Diffusion Models: Towards Immediate Content Removal

**Felix Koulischer**
**IDLab - Ghent University - imec,**
**Ghent, Belgium**

**Johannes Deleu**
**IDLab - Ghent University - imec,**
**Ghent, Belgium**

**Gabriel Raya**
**Jheronimus Academy of Data Science**
**Tilburg University**

**Thomas Demeester***
**IDLab - Ghent University - imec,**
**Ghent, Belgium**

**Luca Ambrogioni***
**Radboud University**
**Donders Institute for Brain,**
**Cognition and Behaviour**

## Abstract

The rise of highly realistic large scale generative diffusion models comes hand in hand wih public safety concerns. In addition to the risk of generating *Not-Safe-For-Work* content from models trained on large internet-scraped datasets, there is a serious concern about reproducing copyrighted material, including celebrity images and artistic styles. We introduce ***Dynamic Negative Guidance*** a theoretically grounded negative guidance scheme that can avoid the generation of unwanted content without drastically harming the diversity of the model. Our approach avoids some of the disadvantages of the widespread, yet theoretically unfounded, Negative Prompting algorithm. Our guidance scheme does not require retraining the conditional model and can therefore be applied as a temporary solution to meet customer requests until model fine-tuning is possible.

## 1 Introduction

Since first proposed as generative models [1, 2, 3, 4], Diffusion models (DMs) have become state-of-the-art models in Text-To-Image (T2I) generation [3, 5]. Recently, models like Midjourney[2] and DALL-E [6] have captured considerable public attention. These models are capable of generating highly realistic images of very diverse sort [7, 8]. These astonishing qualities have also come with their safety concerns. Not only can such models potentially generate *Not Safe For Work* content containing nudity, violence or hateful depictions [9, 10], these models can also memorize copyrighted images [11]. As these large models are trained by scraping data from the internet [10, 12, 13], they can be exposed to copyrighted content, this is highlighted by the fact that models such as Stable Diffusion generate watermarks[14]. Even if the data is free of copyrighted content, diffusion models have further been publicly criticized for replicating artistic styles, or celebrities faces, without the artists or celebrities consent. In practice, filtering datasets of the scale required for the training of T2I models is not feasible in practice, such that removing specific concepts from the outputs of large, mostly black box, model is all but trivial. The most practical solution proposed in the literature consists in fine-tuning specific layers of existing models such as to remove these undesired concepts

---

*Joint Senior Authors

[2]Available at `https://www.midjourney.com/`

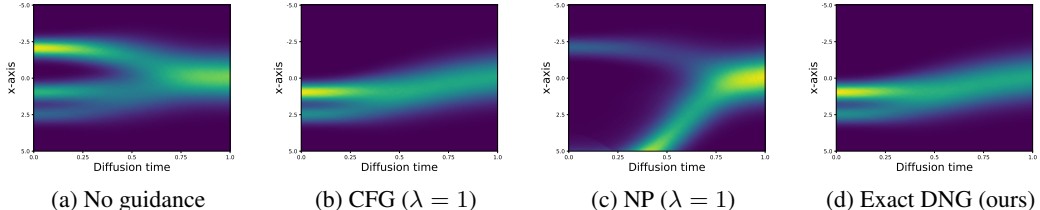

| (a) No guidance | (b) CFG ($\lambda = 1$) | (c) NP ($\lambda = 1$) | (d) Exact DNG (ours) |

Figure 1: Comparison between CFG, NP, and DNG using the exact posterior (all used with $\lambda = 1$). DNG with the exact posterior is equivalent to CFG, while NP fails to sample the target distribution.

[10, 15, 16, 17]. Fine-tuning these models however requires time, especially as it is of paramount importance to verify that the fine-tuning did not harm the general capacities of the model.

To resolve this, we propose a temporary solution to meet user demand in real-time using a novel theoretically grounded negative guidance scheme, coined ***D****ynamic* ***N****egative* ***G****uidance* [3]. Our method can be seen as a replacement for the widespread, yet severely understudied, Negative Prompting (NP) algorithm [9, 18, 19].

Similarly to NP, our method uses the model's own understanding of a concept for the removal [9, 18]. The fundamental flaw of NP is that, regardless of whether a specific feature $c$ is being generated, the guidance impacts the output of the model. Furthermore, as in NP the guidance is present throughout the entire denoising trajectory, it can cause large deviations from the unguided setting. On the contrary, we find that the theoretically optimal negative guidance scale is of dynamic nature. It depends on the probability of a feature being present, i.e. on the posterior $p(c|x)$. Should there be a high likelihood that an undesired feature be absent from the output, the guidance scale would deactivate itself. Mainstream diffusion models lack access to the posterior $p(c|x)$ during inference. To overcome this limitation, we propose a new method that approximates the posterior by tracking the relevant diffusion Markov Chains during the denoising phase.

To analyze the effect of negative guidance, the task of single class removal from an unconditional diffusion model trained on MNIST and CIFAR10 is studied. In this setting, Dynamic Negative Guidance demonstrates superior performance in preserving model diversity when compared to Negative Prompting and Safe Latent Diffusion [9], especially in the high safety regime. To illustrate the flexibility of the approach, DNG is tested in the context of T2I using Stable Diffusion [8]. These proof-of-concept experiments show that DNG is not only capable of editing images on par with NP, but crucially is also capable of deactivating itself when the model is negatively prompted with semantically unrelated text. Such a dynamic property is crucial when using generic negative prompts, such as a celebrities name, as all generated images are potentially affected by its presence. While methods like fine-tuning are essential to guarantee public safety by entirely removing a concept from the network weights, these approaches can take non-negligible amounts of time. In this regard, we believe that the dynamic nature of our approach makes DNG an attractive temporary solution for immediate concept removal in large pretrained models.

Furthermore, as diffusion models, and AI in general, become increasingly central to technological advancements, understanding their inner functioning is of fundamental importance. By building on solid mathematical foundations, our research provides deeper insights into the concept of negative guidance. This fundamental understanding is essential to mitigate potential risks, allowing us to develop strategies that prevent future harm from misuse or unintended model behavior.

## 2  Approaches to concept erasure

In response to rising public concerns, the research community has been actively pursuing techniques that can effectively remove entire concepts from the outputs of diffusion models. Most methods rely on the finetuning of specific layers inside the U-Net, whereby typically the attention layers are

---

[3]Note that we recently submitted a more theoretical contribution introducing Dynamic Negative Guidance, to ICLR'25 (under review until early 2025), which is in line with the workshop submission guidelines. The current manuscript summarizes key points, but puts more emphasis on the potential application of AI safety. Upon acceptance, the theoretical paper will be referenced for further methodological details. For now, key derivations are provided in Appendices A and B.

targeted [10, 13, 15, 17, 20]. The main difficulty beind fine-tuning approaches is to only remove the targeted concepts without harming the rest of the models capacities [17]. An alternative option to avoid the generation of certain features in T2I applications is to instead focus on the textual prompts used to guide the model [16, 21]. By modifying these prompts before using the model, its output can be manipulated. A good overview of the different approaches is described by [22], where the question of how easily such approaches can be bypassed is treated.

Our approach closely aligns with various training-free guidance methods [23, 24, 25, 26], which are designed to enhance safety by steering diffusion models without retraining or fine-tuning. These schemes typically operate like classifier-guided (CG) techniques [27], applying a force field derived from the gradient of a classifier to the unconditional score field. A prominent example is the *Safe Latent Diffusion* (SLD) method proposed by Schramowski et al. [9], which, similarly to Negative Prompting (NP), leverages the model's knowledge of certain concepts to block undesirable content. SLD differs from NP in two key safety-enhancing features: it employs a heuristic, non-constant guidance scale active only when predictions from the negatively and positively prompted models overlap, and it applies pixel-wise guidance to selectively blend allowed and forbidden content. We find that despite intuitive, this feature is not present in the exact formulas obtained with known score functions (see our derivations in Sec. 3.2). Finally, Chen et al. [11] use a very similar approach to attack the problem of memorization of training samples in DMs, a context in which our scheme could also potentially be used.

## 3 Theory

To derive our theoretically grounded Dynamic Negative Guidance scheme, a thorough understanding of well-known guidance methods such as Classifier-Free guidance (CFG) and Negative Prompting (NP) is necessary. These are first briefly introduced, after which the proposed scheme is presented with a way to approximate the posterior by tracking the likelihoods of the relevant Markov Chains. All the derivations, and their implications, are described in a full paper, sketches of how the underlying theory is derived are provided in Appendix A and B.

### 3.1 Classifier Free Guidance and Negative Prompting

Diffusion Models are score-based models, and therefore never explicitly model the underlying distributions. Instead, they restrict themselves to learning the score of a distribution, defined as the gradient of its log likelihood $\nabla_{\boldsymbol{x}} \log p_t(\boldsymbol{x})$. As most tasks require conditional generation, it is more common to sample from a sharpened conditional distribution $p_t(\boldsymbol{x}|\boldsymbol{c}) \propto p_t(\boldsymbol{x})p_t(c|\boldsymbol{x})^\lambda$ [27]. From a score based perspective, this corresponds to adding a guidance field defined through the gradient of a posterior $\nabla_{\boldsymbol{x}} \log p_t(c|\boldsymbol{x})$. During most of the denoising process, the state $\boldsymbol{x}$ contains large amounts of noise, making such classification impractical. To overcome this issue, Ho et al. [7] proposed retaining the guidance scale $\lambda$ while still training the joint conditional model, and rewriting the posterior using Bayes' rule: $p_t(c|\boldsymbol{x}) \propto p_t(\boldsymbol{x}|c)/p_t(\boldsymbol{x})$. From a score-based perspective, this results in the well-known Classifier-Free Guidance [7] equation:

$$\nabla_{\boldsymbol{x}} \log p_t(\boldsymbol{x}|\boldsymbol{c}) = \nabla_{\boldsymbol{x}} \log p_t(\boldsymbol{x}) + \lambda\big(\nabla_{\boldsymbol{x}} \log p_t(\boldsymbol{x}, \boldsymbol{c}) - \nabla_{\boldsymbol{x}} \log p_t(\boldsymbol{x})\big) \tag{1}$$

It soon became clear that by reverting the sign of the guidance scale $\lambda$ a repulsive guidance was obtained, this became known as *Negative Prompting*[4]. Despite being widely accepted in the diffusion community, NP is fundamentally flawed. To understand why, it suffices to understand that the guidance field defined by $\nabla_{\boldsymbol{x}} \log p_t(\boldsymbol{x}, \boldsymbol{c}) - \nabla_{\boldsymbol{x}} \log p_t(\boldsymbol{x})$ in Eq. (1) is strongest in the regions where $\boldsymbol{x}$ is furthest from $\boldsymbol{c}$. By simply inverting the field's direction, regions unrelated to the undesired feature $\boldsymbol{c}$ would receive much stronger guidance than those actually related to $\boldsymbol{c}$. This flaw is particularly apparent in one dimension, as visualized in Figure 1c, in which NP completely fails to sample the correct target distribution. Conditioning variables and scores associated with an undesired (or negative) condition are denoted in red ($c_-$ and $s_{\theta,c_-}$, respectively). Those referring to wanted (i.e., positive) prompts are written in green ($c_+$ and $s_{\theta,c_+}$).

---

[4]This implies sampling from $p_t(\boldsymbol{x}|\boldsymbol{c}) \propto p_t(\boldsymbol{x})/p_t(\boldsymbol{c}|\boldsymbol{x})^\lambda$

## 3.2 Dynamic Negative Guidance

A well-defined negative guidance scheme can be derived by realizing that the desired posterior can be expressed as the opposite of the undesired posterior, i.e. $p_t(\boldsymbol{x}|\boldsymbol{c}_+) = 1 - p_t(\boldsymbol{x}|\boldsymbol{c}_-)$. Sampling from this conditional distribution, with the posterior emphasized by a positive exponent $\lambda_0$ similar to the guidance scale, happens through the following score:

$$
\begin{aligned}
\nabla_{\boldsymbol{x}} \log p_t(\boldsymbol{x}|\boldsymbol{c}_+) &= \nabla_{\boldsymbol{x}} \log p_t(\boldsymbol{x}) + \lambda_0 \nabla_{\boldsymbol{x}} \log \big(1 - p_t(\boldsymbol{c}_-|\boldsymbol{x})\big) \\
&= \nabla_{\boldsymbol{x}} \log p_t(\boldsymbol{x}) - \lambda_0 \frac{p_t(\boldsymbol{c}_-|\boldsymbol{x})}{1 - p_t(\boldsymbol{c}_-|\boldsymbol{x})} \big(\nabla_{\boldsymbol{x}} \log p_t(\boldsymbol{x}|\boldsymbol{c}_-) - \nabla_{\boldsymbol{x}} \log p_t(\boldsymbol{x})\big) \quad (2) \\
&= \nabla_{\boldsymbol{x}} \log p_t(\boldsymbol{x}) - \lambda(\boldsymbol{x},t)\big(\nabla_{\boldsymbol{x}} \log p_t(\boldsymbol{x}|\boldsymbol{c}_-) - \nabla_{\boldsymbol{x}} \log p_t(\boldsymbol{x})\big)
\end{aligned}
$$

This equation reveals that the guidance scale should be *dynamically* rescaled throughout the denoising. The guidance scale becomes asymptotically large as $p_t(\boldsymbol{c}_-|\boldsymbol{x}) \to 1$, while it remains small when $p_t(\boldsymbol{c}_-|\boldsymbol{x}) \to 0$. These equations are validated in the one dimensional setting in which the posterior is explicitly available. This can be seen in Figure 1 on which it is visible that applying DNG with a repulsive force directed away from the undesired mode is equivalent to applying CFG with an attractive force towards the desired modes.

Diffusion models being score-based models do not give the possibility of easily computing the likelihoods. Inspired by recent work of Li et al. [28] that recognizes DMs as zero-shot classifiers, we propose to track the likelihoods of the conditional and unconditional Markov Chains throughout the diffusion process to recompose the posterior. Further, realizing that both $p_t(\boldsymbol{x}_t|\boldsymbol{x}_{t+1}, \boldsymbol{c}_-; \theta)$ and $p_t(\boldsymbol{x}_t|\boldsymbol{x}_{t+1}; \theta)$ are Gaussian of same variance with modelled means $\boldsymbol{\mu}_{t,\theta}(\boldsymbol{x}_{t+1}|\boldsymbol{c}_-)$ and $\boldsymbol{\mu}_{t,\theta}(\boldsymbol{x}_{t+1})$ allows us to summarize the approach as:

$$
\begin{aligned}
\log p_t(\boldsymbol{c}_-|\boldsymbol{x}_{t-1}) &\simeq \log p_t(\boldsymbol{c}_-|\boldsymbol{x}_t) \\
&= \log p_t(\boldsymbol{c}_-) + \sum_{i=T}^{t} \big(\log p_{i-1}(\boldsymbol{x}_{i-1}|\boldsymbol{x}_i, \boldsymbol{c}_-; \theta) - \log p_{i-1}(\boldsymbol{x}_{i-1}|\boldsymbol{x}_i; \theta)\big) \\
&= \log p_{t+1}(\boldsymbol{c}_-|\boldsymbol{x}_{t+1}) + \big(\log p_t(\boldsymbol{x}_t|\boldsymbol{x}_{t+1}, \boldsymbol{c}_-; \theta) - \log p_t(\boldsymbol{x}_t|\boldsymbol{x}_{t+1}; \theta)\big) \\
&= \log p_{t+1}(\boldsymbol{c}_-|\boldsymbol{x}_{t+1}) - \frac{1}{2\sigma_{t+1}^2}\big(\|\boldsymbol{x}_t - \boldsymbol{\mu}_{t,\theta}(\boldsymbol{x}_{t+1}|\boldsymbol{c}_-)\|^2 - \|\boldsymbol{x}_t - \boldsymbol{\mu}_{t,\theta}(\boldsymbol{x}_{t+1})\|^2\big)
\end{aligned}
$$
$$(3)$$

The often used assumption of an infinitesimal diffusion process for which $\log p_{t-1} \simeq \log p_t$ is required to avoid an implicitly defined scheme. The term added to the posterior in Eq. (10) can be positive or negative, respectively corresponding to an increase or a decrease of the posterior likelihood, and by extension thereof, of the guidance scale. To regularize this dynamic posterior estimation, we propose adding a linear transformation before the difference of Euclidean distances. Rescaling the difference by a factor $\tau \in\ ]0, 1[$ can diminish stochastic fluctuations present during denoising, while a small offset $\delta$ creates a slight bias towards increasing the posterior. Should an allowed image being generated, this offset is completely dominated by the very large difference.

## 4 Experiments

### 4.1 Class removal

To analyze the invasiveness of our proposed method, the different guidance schemes are compared in the context of image generation on labelled datasets, in the present case MNIST and CIFAR10 are considered. The objective is to avoid generating one of the classes by guiding an unconditional model with a model trained solely on that specific class. The *safety* of the approach is quantified by a classifier that assesses the percentage of generated images that belong to the forbidden class, while the *diversity* is measured by examining the overall distribution of generated classes across all images. This distribution ideally contains a single zero and equal weight on all other classes (see Figure 2c). To measure how this ideal case is approximated, the KL-divergence between ideal and generated distributions is computed. The *quality* of the model is measured through the standard FID metric computed between 10420 generated images and the training data excluding the undesired class [29][5].

---

[5]As a statistical metric the FID is not only negatively affected by poor quality generation, but also by large class imbalances.

Both the FID and the KL-divergence can be compared for different approaches at equal safety. To vary the safety of different approaches a sweep over the initial guidance scale $\lambda_0$ is performed. These graphs are shown for MNIST in Fig. 2a and for CIFAR10 in Fig. 2b. In the regime of high safety, where as few as possible forbidden images are generated, DNG significantly outperforms concurrent approaches, showcasing that the image removal performed by DNG is less invasive.

## 4.2 Illustrative Results in T2I

While previous experiments showcase the capacities of DNG, it is impractical to train a network on solely forbidden content. In the case of celebrity removal, this would imply requiring a model only capable of generating a single celebrity, for instance "Taylor Swift" before it can be removed. Instead, our framework solely requires a conditional model, a prerequisite met by all T2I models. In this setting, the model's knowledge of certain concept such as "Taylor Swift" can be used to our own advantage, by prompting the model with the name of the celebrity such a specialized negative model is obtained. As proof of concept that the proposed scheme remains sensible in this setting, we prompt Stable Diffusion 2 [9] to generate an image of "Taylor Swift riding a horse" and then add a negative prompt containing "Taylor Swift" (visible in Figure 3 (e)-(f)). These preliminary experiments demonstrate that when the hyperparameters of DNG are correctly tuned, our approach is just as efficient as NP at removing visible features. On the other hand, we also show that when generating an image completely unrelated to "Taylor Swift", such as for instance "An English breakfast", the dynamic guidance scale defined by DNG falls to zero, leaving the images close to unaltered (visible in Figure 3 (a)-(c)). This is not at all the case when using NP, which is just as strong regardless whether the undesired feature is present or not in the image.

As already highlighted in the literature [5, 25, 30, 31], the generation of images happens in phases. The main advantage of our self-regulated guidance scale is that such events can be observed by tracking the posterior, or equivalently, the guidance scale. This is visualized in Figure 4. The red line corresponds to semantically unrelated negative guidance (being "A truck" when generating "Taylor Swift riding a horse", and "Taylor Swift" when generating "An English breakfast") for which it is visible that the guidance is efficiently deactivated. The green line corresponds to semantically related negative guidance (being "Taylor Swift" when generating "Taylor Swift riding a horse", and "An egg" when generating "An English breakfast") for which it is visible that the guidance is active.

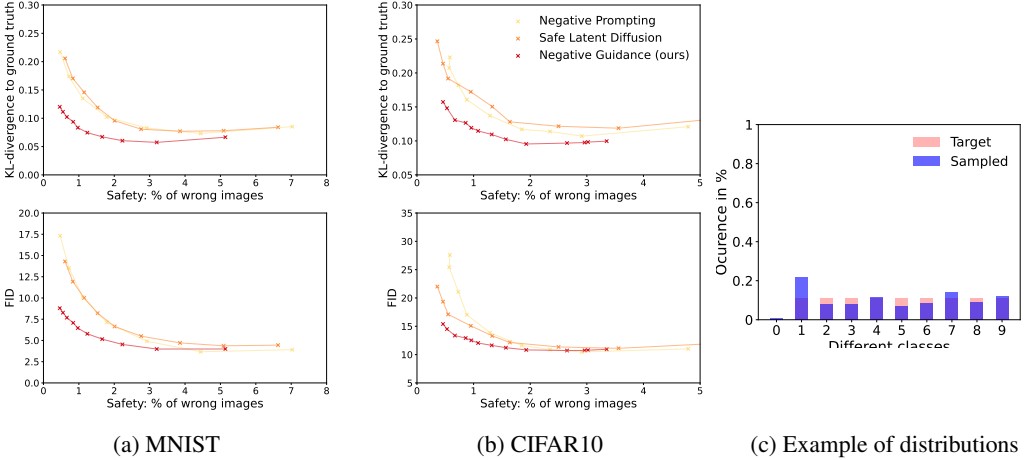

(a) MNIST        (b) CIFAR10        (c) Example of distributions

Figure 2: Comparison of diversity (top) and quality (bottom) as a function of safety for SLD, NP and DNG (ours) when removing a specific class measured respectively using the KL-divergence and the FID. To reduce the percentage of undesired images generated, the initial guidance scale is increased. In Fig. 2a the number zero is removed. In Fig. 2b the class *airplane* is removed. An example showing how the KL-divergence is computed is shown in Fig. 2c, the example is taken using our approach on MNIST. The class corresponding to the number one is oversampled, as it lies furthest from the undesired class zero.

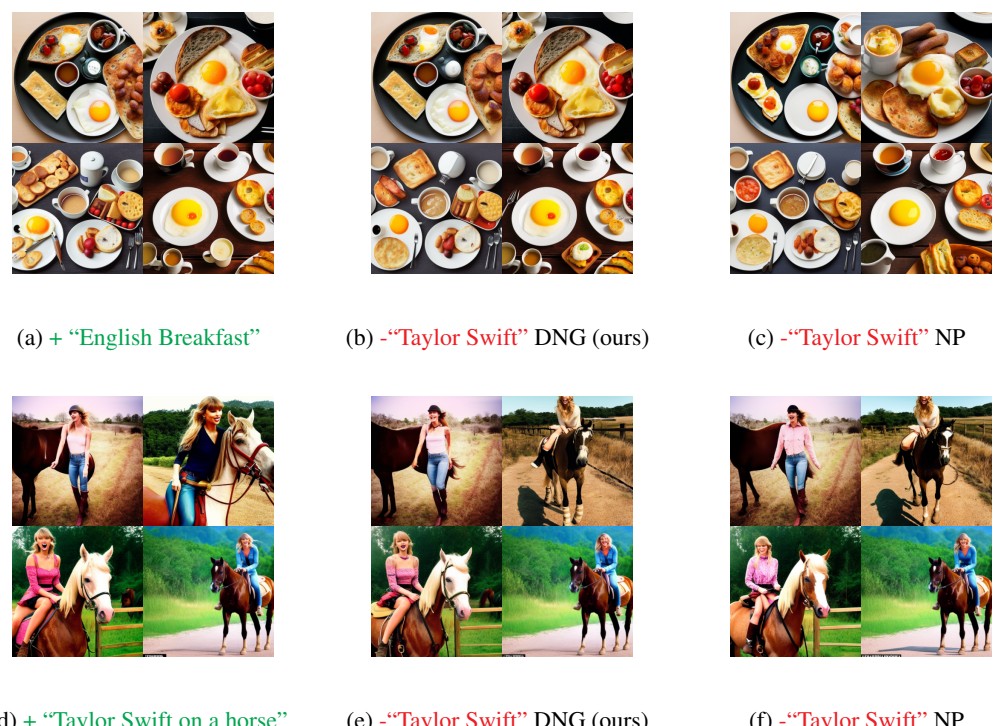

(a) + "English Breakfast"     (b) -"Taylor Swift" DNG (ours)     (c) -"Taylor Swift" NP

(d) + "Taylor Swift on a horse"     (e) -"Taylor Swift" DNG (ours)     (f) -"Taylor Swift" NP

Figure 3: Examples illustrating that our DNG scheme keeps the model diversity. The guidance is deactivated in the case of an unrelated positive prompt (illustrated in (a)-(c)), while still capable of removing the celebrity's identity should it be present in the generated images (illustrated in (e)-(f))

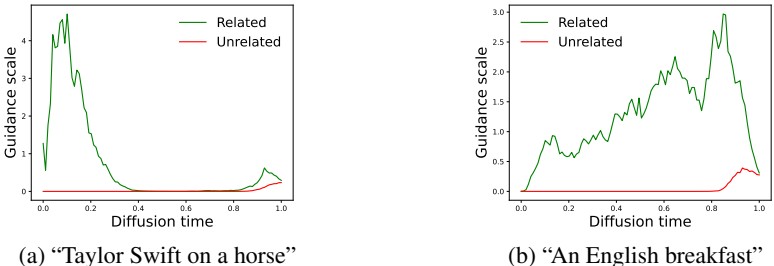

(a) "Taylor Swift on a horse"     (b) "An English breakfast"

Figure 4: Illustration of our *dynamic* guidance scale. For semantically unrelated guidance, such as "Taylor Swift" for "An English breakfast", the guidance scale drops to zero. While for of semantically related negative guidance, such as "Taylor Swift" for "Taylor Swift riding a horse" the guidance is activated.

## 5  Conclusion

We presented Dynamic Negative Guidance, a novel scheme for negative guidance of diffusion models, which overcomes fundamental limitations of the popular Negative Prompting approach. Our method better preserves the diversity of underlying models in the context of single class removal. Crucially, the dynamic nature of the guidance scale allows our method to switch off automatically when the undesired feature defined by the negative prompt is not present. Our method could be used as a temporary but immediate solution (i.e., without requiring any diffusion model retraining), for example to comply with demands from public figures or artists requiring the removal of a model's ability to reproduce their appearance or style, even when prompted to do so by users.

**Acknowledgments**

This research was partly funded by the Research Foundation - Flanders (FWO-Vlaanderen) under grant G0C2723N and by the Flemish Government (AI Research Program). Gabriel Raya was funded by the Dutch Research Council (NWO) as part of the CERTIF-AI project (file number 17998).

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

---

**Algorithm 1** Dynamic Negative Guidance

---

**Input:** Pre-trained unconditional DDPM with noise prediction $\epsilon_\theta$, Pre-trained *to-forget* DDPM with noise prediction $\epsilon_{f,\theta}$, guidance scale $\lambda_0$, prior $p_0$ and Temperature $\tau$

$z \sim \mathcal{N}(0, \boldsymbol{I})$, $\boldsymbol{x}_T \sim \mathcal{N}(0, \boldsymbol{I})$

$p(\boldsymbol{c}\text{-}|\boldsymbol{x}_T) = p_0$          `Initialize posterior and guidance scale`

$\boldsymbol{\lambda_T}(\boldsymbol{x}_T) = \lambda_0 \frac{p(\boldsymbol{c}\text{-}|\boldsymbol{x}_T)}{1-p(\boldsymbol{c}\text{-}|\boldsymbol{x}_T)}$

**for** $t = T, \ldots, 1$ **do**

    $\epsilon_{\theta,\text{guid}}(\boldsymbol{x}_t) = \epsilon_\theta(\boldsymbol{x}_t) - \lambda_t(\boldsymbol{x}_t)\big(\epsilon_{f,\theta}(\boldsymbol{x}_t) - \epsilon_\theta(\boldsymbol{x}_t)\big)$          `Apply guidance`

    $\boldsymbol{x}_{t-1} = \frac{1}{\sqrt{\alpha_t}}\big(\boldsymbol{x}_t - \frac{1-\alpha_t}{\sqrt{1-\bar{\alpha}_t}}\epsilon_{\theta,\text{guid}}(\boldsymbol{x}_t)\big) + \sqrt{\beta_t}\boldsymbol{z}$          `DDPM Step`

    $p(\boldsymbol{c}\text{-}|\boldsymbol{x}_{t-1}) = \text{Compute posterior}\big(p(\boldsymbol{c}\text{-}|\boldsymbol{x}_t), \boldsymbol{x}_{t-1}, \boldsymbol{x}_t, \epsilon_\theta(\boldsymbol{x}_t), \epsilon_{f,\theta}(\boldsymbol{x}_t)\big)$          `See Algorithm 2`

    $\lambda_t(\boldsymbol{x}_{t-1}) = \lambda_0 \frac{p(\boldsymbol{c}\text{-}|\boldsymbol{x}_{t-1})}{1-p(\boldsymbol{c}\text{-}|\boldsymbol{x}_{t-1})}$          `Compute new guidance scale`

**end for**

---

## A    Derivation of Dynamic Negative Guidance

The objective of our DNG scheme is to first transform the undesired condition $\boldsymbol{c}\text{-}$ into an analogue desired condition $\boldsymbol{c}_+$, where the latter should correspond to "anything *but* $\boldsymbol{c}\text{-}$. This is achieved by considering $p(\boldsymbol{c}_+|\boldsymbol{x}) = 1 - p(\boldsymbol{c}\text{-}|\boldsymbol{x})$. By using Bayes rule the conditional can be rewritten as:

$$p_t(\boldsymbol{x}|\boldsymbol{c}_+) \propto p_t(\boldsymbol{x})\big(1 - p_t(\boldsymbol{c}\text{-}|\boldsymbol{x})\big) \tag{4}$$

From a score-based perspective this gives:

$$\nabla_{\boldsymbol{x}} \log p_t(\boldsymbol{x}|\boldsymbol{c}_+) = \nabla_{\boldsymbol{x}} \log p_t(\boldsymbol{x}) + \nabla_{\boldsymbol{x}} \log \big(1 - p_t(\boldsymbol{c}\text{-}|\boldsymbol{x})\big) \tag{5}$$

Unlike in CFG, the last term is not directly recognizable as a linear combination of scores. By using the chain rule to remove the logarithm and then artificially reintroducing it, a typical score function is obtained:

$$
\begin{aligned}
\nabla_{\boldsymbol{x}} \log \big(1 - p_t(\boldsymbol{c}\text{-}|\boldsymbol{x})\big) &= -\frac{1}{1 - p_t(\boldsymbol{c}\text{-}|\boldsymbol{x})} \nabla_{\boldsymbol{x}} p_t(\boldsymbol{c}\text{-}|\boldsymbol{x}) \\
&= -\frac{p_t(\boldsymbol{c}\text{-}|\boldsymbol{x})}{1 - p_t(\boldsymbol{c}\text{-}|\boldsymbol{x})} \frac{1}{p_t(\boldsymbol{c}\text{-}|\boldsymbol{x})} \nabla_{\boldsymbol{x}} p(\boldsymbol{c}\text{-}|\boldsymbol{x}) \\
&= -\frac{p_t(\boldsymbol{c}\text{-}|\boldsymbol{x})}{1 - p_t(\boldsymbol{c}\text{-}|\boldsymbol{x})} \nabla_{\boldsymbol{x}} \log p_t(\boldsymbol{c}\text{-}|\boldsymbol{x})
\end{aligned}
\tag{6}
$$

By proceeding in a fashion very similar to CFG [7], i.e. rescaling the posterior by a positive exponent $\lambda_0$ (analogous to the guidance scale) and rewriting the posterior as a linear combination of a conditional and unconditional model, one obtains:

$$
\begin{aligned}
\nabla_{\boldsymbol{x}} \log p_t(\boldsymbol{x}|\boldsymbol{c}_+) &= \nabla_{\boldsymbol{x}} \log p_t(\boldsymbol{x}) - \lambda_0 \frac{p_t(\boldsymbol{c}\text{-}|\boldsymbol{x})}{1 - p_t(\boldsymbol{c}\text{-}|\boldsymbol{x})} \big(\nabla_{\boldsymbol{x}} \log p_t(\boldsymbol{x}|\boldsymbol{c}\text{-}) - \nabla_{\boldsymbol{x}} \log p_t(\boldsymbol{x})\big) \\
&= \nabla_{\boldsymbol{x}} \log p_t(\boldsymbol{x}) - \lambda(\boldsymbol{x}, t)\big(\nabla_{\boldsymbol{x}} \log p_t(\boldsymbol{x}|\boldsymbol{c}\text{-}) - \nabla_{\boldsymbol{x}} \log p_t(\boldsymbol{x})\big)
\end{aligned}
\tag{7}
$$

This is the fundamental equation behind our Dynamic Negative Guidance scheme. Its dynamic nature is apparent from the time- and state-dependence of the guidance scale $\lambda(\boldsymbol{x}, t)$. On the contrary to NP, a theoretically optimal negative guidance scheme is only active in regions related to $\boldsymbol{c}\text{-}$, i.e. when $p(\boldsymbol{c}\text{-}|\boldsymbol{x}) \to 1$. Algorithm 1 summarizes the DNG scheme.

## B    Estimation of the posterior

Diffusion models, being score-based, do not directly offer the underlying distributions. We however find that the posterior $p(\boldsymbol{c}|\boldsymbol{x})$ can be estimated by tracking the Markov chains of the conditional and unconditional models, defined by $p(\boldsymbol{x}|\boldsymbol{c}) = p(\boldsymbol{x}_T|\boldsymbol{c}) \prod_{i=t+1}^{T} p_i(\boldsymbol{x}_{i-1}|\boldsymbol{x}_i, |\boldsymbol{c}; \theta)$ and $p(\boldsymbol{x}) = p(\boldsymbol{x}_T) \prod_{i=t+1}^{T} p_i(\boldsymbol{x}_{i-1}|\boldsymbol{x}_i; \theta)$.

The posterior is then recognizable as:

$$p_t(\boldsymbol{c}|\boldsymbol{x}_{t:T}) = p(\boldsymbol{c})\frac{p(\boldsymbol{x}_{t:T}|\boldsymbol{c})}{p(\boldsymbol{x}_{t:T})}$$

$$\iff \log p_t(\boldsymbol{c}|\boldsymbol{x}_{t:T}) = \log p(\boldsymbol{c}) + \sum_{i=T}^{t+1}\big(\log p_{i-1}(\boldsymbol{x}_{i-1}|\boldsymbol{x}_i,\boldsymbol{c};\theta) - \log p_{i-1}(\boldsymbol{x}_{i-1}|\boldsymbol{x}_i;\theta)\big) \quad (8)$$

$$= \log p_{t+1}(\boldsymbol{c}|\boldsymbol{x}_{t+1:T}) + \big(\log p_t(\boldsymbol{x}_t|\boldsymbol{x}_{t+1},\boldsymbol{c};\theta) - \log p_t(\boldsymbol{x}_t|\boldsymbol{x}_{t+1};\theta)\big)$$

As all transition are approximately Gaussians with mean $\boldsymbol{\mu}_{\theta,t}(\boldsymbol{x})$ and variance $\sigma_t$, the last equation leads to an iterative update rule:

$$\log p_t(\boldsymbol{c}|\boldsymbol{x}_t) = \log p_{t+1}(\boldsymbol{c}|\boldsymbol{x}_{t+1}) - \frac{1}{2\sigma_t^2}\big(\|\boldsymbol{x}_t - \boldsymbol{\mu}_{t,\theta}(\boldsymbol{x}_{t+1}|\boldsymbol{c})\|^2 - \|\boldsymbol{x}_t - \boldsymbol{\mu}_{t,\theta}(\boldsymbol{x}_{t+1})\|^2\big) \quad (9)$$

To know in which point $\boldsymbol{x}_t$ the posterior needs to be estimated, the guidance scale is required, which depends itself on the posterior. An implicit problem is therefore defined. To resolve the implicitness of the above equation, we propose to assume that the posterior changes slowly such that the guidance scale can, up to first order, be approximated by its previous value. This assumption becomes exact as the number of diffusion time steps becomes infinitely large, an assumption often used in the diffusion literature. In essence, the computation of the guidance scale and that of the denoising is staggered in time. To obtain the guidance scale required to find $\boldsymbol{x}_{t-1}$, the posterior at time step $t$ is used, which solely depends on $\boldsymbol{x}_t$, $\boldsymbol{\mu}_{t,\theta}(\boldsymbol{x}_{t+1})$ and $\boldsymbol{\mu}_{t,\theta}(\boldsymbol{x}_{t+1}|\boldsymbol{c})$, which are all known. The approximation can be described mathematically as:

$$\log p_{c,t-1} \simeq \log p_{c,t}$$

$$= \log p_{c,t+1} - \frac{1}{2\sigma_{t+1}^2}\big(\|\boldsymbol{x}_t - \boldsymbol{\mu}_{t,\theta}(\boldsymbol{x}_{t+1}|\boldsymbol{c})\|^2 - \|\boldsymbol{x}_t - \boldsymbol{\mu}_{t,\theta}(\boldsymbol{x}_{t+1}))\|^2\big) \quad (10)$$

In the case treated in this work, the conditional model is defined by the udesired condition $\boldsymbol{c}_-$. Adding a regularizing linear transformation, parametrized by $\tau$ and $\delta$, to the difference of Euclidean distances leads to a scheme defined by Algorithm 2.

---

**Algorithm 2** Compute posterior

---

**Input:** Previous estimation of filtering posterior $p_t(\boldsymbol{c}_-|\boldsymbol{x}_t)$, Updated noisy state $\boldsymbol{x}_{t-1}$, previous noisy state $\boldsymbol{x}_t$, unconditional noise prediction $\epsilon_\theta(\boldsymbol{x}_t)$, *to-forget* noise prediction $\epsilon_{f,\theta}(\boldsymbol{x}_t)$, diffusion constants $\alpha_t$, $\bar{\alpha}_t$, prior $p_0$, Temperature $\tau$, offset $\delta$, minimal and maximal posterior values $p_{\min}$ and $p_{\max}$

$\sigma_t^2 = 1 - \alpha_t$                                         `Variance of Gaussian at` $t$

$\boldsymbol{\mu}(\boldsymbol{x}_t) = \frac{1}{\sqrt{\alpha_t}}\big(\boldsymbol{x}_t - \frac{1-\alpha_t}{\sqrt{1-\bar{\alpha}_t}}\epsilon_{f,\theta}(\boldsymbol{x}_t)\big)$             `Undesired` `mean prediction`

$\boldsymbol{\mu}(\boldsymbol{x}_t) = \frac{1}{\sqrt{\alpha_t}}\big(\boldsymbol{x}_t - \frac{1-\alpha_t}{\sqrt{1-\bar{\alpha}_t}}\epsilon_\theta(\boldsymbol{x}_t)\big)$              `Unconditional mean prediction`

$p(\boldsymbol{c}_-|\boldsymbol{x}_{t-1}) = p_t(\boldsymbol{c}_-|\boldsymbol{x}_t)\exp\Big(-\frac{\tau}{2\sigma_t^2}\big(\|\boldsymbol{x}_{t-1} - \boldsymbol{\mu}(\boldsymbol{x}_t)\|^2 - \|\boldsymbol{x}_{t-1} - \boldsymbol{\mu}(\boldsymbol{x}_t)\|^2\big) + \frac{\delta}{2\sigma_t^2}\Big)$

$p(\boldsymbol{c}_-|\boldsymbol{x}_{t-1}) = \text{Clamp}\big(p(\boldsymbol{c}_-|\boldsymbol{x}_{t-1}), \min = p_{\min}, \max = p_{\max}\big)$

**Output:** Approximate posterior probability $p(\boldsymbol{c}_-|\boldsymbol{x}_{t-1})$

---

## C   Experimental details

### C.1   DNG hyperparameters

The hyperparameters introduced in the framework of DNG, being the prior $p(c)$, the temperature $\tau$ and the bias $\delta$ all have distinct effects. The prior dictates the initial guess for the posterior, i.e. $p_T(c|\boldsymbol{x}_T) = p(c)$ and therefore also the initial guidance scale $\lambda(\boldsymbol{x},T) = \lambda\frac{p(c)}{1-p(c)}$. Choosing a low prior can ensure that negative guidance is not immediately active. The temperature hyperparameter $\tau$ can help to reduce the fluctuations caused by the stochastic denoising process, we find that a value of around $\tau = 0.2$ works well in practice. The offset hyperparameter $\delta$ gives the model a slight bias towards increasing the prior, it could be alternatively described as making the prior time dependent.

From a practical point of view, it's consequence is that when both the positively and negatively prompted models give similar prediction, the posterior increases faster, a highly desirable property. In practice, we suggest either choosing a relatively large prior with no offset (such as we have done for the MNIST experiments), or a low prior with a small offset (such as we have done for the CIFAR10 and Stable Diffusion experiments). We find that choosing an offset two, or even three, orders of magnitude lower than the temperature delivers satisfactory results.

## C.2 Details for class removal experiments

For the class removal experiments, unconditional MNIST and CIFAR10 diffusion models are required. For MNIST our own model is trained, while for CIFAR10 the pretrained model from [2] is used[6]. For both datasets, a model trained on solely one of the classes is required. For this, our own models are trained on all the *zeros* of MNIST and all the *airplanes* of CIFAR10. Notice that all the guidance approaches are compared using the same two networks, reducing any additional bias due to the choices of network architectures. To analyze the generated images, a vision classifier is required. For MNIST our own basic convolutional based classifier is trained, obtaining over 98% accuracy over a test set. For CIFAR10, a pretrained vision transformer classifier is used[7] [32].

Hyperparameter values for our Negative Guidance scheme for the different datasets are given in Table 1. A discussion explaining the choice of the different hyperparameters of our scheme is included in C.1.

For Safe Latent Diffusion [9] a hyperparameter search was performed to obtain the values that perform best at high safety. The most important hyperparameter is the threshold value at which guidance is activated. We found that even in the setting of MNIST and CIFAR (quite far from the setting of Stable Diffusion in which the scheme is proposed), a threshold value of $\lambda_{\text{thresh}} = 0.04$ still performs optimally at high safety. This displays the flexibility of the approach proposed by [9]. The other hyperparameters are chosen as follows: $s_s = 100$, $\beta_m = 0.2$, $s_m = 0.1$.

| Dataset | Prior $p(c)$ | Temperature $\tau$ | Offset $\delta$ |
|---------|--------------|--------------------|-----------------|
| MNIST   | 0.25         | 0.25               | 0.0             |
| CIFAR10 | 0.01         | 0.2                | 0.0002          |

Table 1: The prompt specific hyperparameters chosen for our Dynamic Negative Prompting.

The negative guidance scale has to be chosen significantly differently for the various approaches. While in SLD the guidance scale can only be smaller or equal to the initial guidance scale $\lambda_0$, our scheme considers a dynamic self-regulated guidance, which can therefore require very different initial values. In practice, the guidance scale used in DNG is chosen one order of magnitude larger than that of NP. For NP or SLD, we chose values of around $0.5$, while for DNG we chose values around $5$.

---

[6]The pretrained model can be downloaded from huggingface at `https://huggingface.co/google/ddpm-cifar10-32`

[7]The pretrained classifier can be downloaded from Hugging Face at `https://huggingface.co/aaraki/vit-base-patch16-224-in21k-finetuned-cifar10`

