# OpenReview forum: "Dynamic Negative Guidance of Diffusion Models: Towards Immediate Content Removal"
_NeurIPS.cc/2024/Workshop/SafeGenAi — SafeGenAi Poster_

### Official Review · Reviewer_no4t · 2024-10-09
**Motivation and approach are well written and experimental results are good.**

**Rating:** 7
**Confidence:** 3

**Review:**

This paper proposes a method (Dynamic Negative Guidance) to guide the diffusion model in avoiding outputting sensitive images.
Figure 2 shows a good result compared to the baseline.
The motivation is also clearly stated in the introduction, which I think is appropriate for a workshop.
However, I am not familiar with the diffusion model, so please do not emphasize my review too much.

---

### Official Review · Reviewer_19xd · 2024-10-12
**machine unlearning without fine-tuning (via negative guidance)**

**Rating:** 6
**Confidence:** 4

**Review:**

This paper attacks the problem of machine unlearning in diffusion models. In particular, the authors propose a solution when fine-tuning is not acceptable, and the only option is a proper guidance (far from the "forgetting regions"). In short, they propose using dynamic negative guidance.
The paper is simple, but interesting from the theoretical perspective. However, the method itself (in the presented evaluation setting) seems to be only slightly better than negative prompting and safe latent diffusion. For example, the results in Fig. 3 ((d) - (f)) are unconvincing. I wouldn't say they are significantly different from NP method.
Overall, I think that the paper still needs some work on the experimental setting and justification for using this model.